# Exosomal CD40, CD25, and Serum CA19-9 as Combinatory Novel Liquid Biopsy Biomarker for the Diagnosis and Prognosis of Patients with Pancreatic Ductal Adenocarcinoma

**DOI:** 10.3390/ijms26041500

**Published:** 2025-02-11

**Authors:** Paul David, Dina Kouhestani, Frederik J. Hansen, Sushmita Paul, Franziska Czubayko, Alara Karabiber, Nadine Weisel, Bettina Klösch, Susanne Merkel, Jan Ole-Baur, Andreas Gießl, Jan Van Deun, Julio Vera, Anke Mittelstädt, Georg F. Weber

**Affiliations:** 1Department of Surgery, University Hospital Erlangen, 91054 Erlangen, Germany; paul.david@uk-erlangen.de (P.D.); dina.kouhestani@ibbnetzwerk-gmbh.com (D.K.); frederik.hansen@med.uni-duesseldorf.de (F.J.H.); franziska.czubayko@uk-erlangen.de (F.C.); alara.karabiber@uk-erlangen.de (A.K.); nadine.weisel@uk-erlangen.de (N.W.); bettina.kloesch@uk-erlangen.de (B.K.); susanne.merkel@uk-erlangen.de (S.M.); anke.mittelstaedt@uk-erlangen.de (A.M.); 2Department of Dermatology, University Hospital Erlangen, 91054 Erlangen, Germany; sushmita.paul@uk-erlangen.de (S.P.); jan-ole.baur@klinikum-bayreuth.de (J.O.-B.); julio.vera-gonzalez@uk-erlangen.de (J.V.); 3Medizinische Klinik IV (Hämatologie und Onkologie), Klinikum Bayreuth GmbH, 95445 Bayreuth, Germany; 4Department of Ophthalmology, University Hospital Erlangen, 91054 Erlangen, Germany; andreas.giessl@uk-erlangen.de; 5Deutsches Zentrum für Immuntherapie, Friedrich-Alexander-Universität Erlangen-Nürnberg and Universitätsklinikum Erlangen, 91054 Erlangen, Germany; 6Bavarian Cancer Research Center (BZKF), 91052 Erlangen, Germany

**Keywords:** exosomes, liquid biopsy, non-invasive biomarker, PDAC

## Abstract

The poor prognosis of pancreatic ductal adenocarcinoma (PDAC) is largely due to several challenges, such as late diagnosis, early metastasis, limited response to chemotherapy, aggressive tumor biology, and high rates of tumor recurrence. Therefore, the development of a non-invasive and effective method for early detection of PDAC is crucial to improving patient outcomes. Continued research and exploration in this area are essential to enhance early detection methods and ultimately improve the prognosis for individuals with PDAC. In this study, we examined 37 exosomal surface proteins through a multiplex flow cytometry test on peripheral plasma samples from a group of 51 clinical control individuals (including healthy volunteers and non-cancer patients (Cholecystectomy, Hernia, healthy volunteers)), 21 pancreatitis, and 48 patients diagnosed with PDAC. Our research findings revealed that the level of exosomal CD40 expression is significantly lower in patients with PDAC and pancreatitis compared to non-cancer patients (*p* < 0.0001). Additionally, pancreatitis patients exhibited higher levels of exosomal CD25 expression than PDAC patients (*p* = 0.0104). PDAC patients with higher exo-CD40 had worse survival than patients with lower exo-CD40 (*p* = 0.0035). Similarly, PDAC patients with higher exo-CD25 showed worse survival in comparison to patients with lower exo-CD25 (*p* = 0.04). Statistical analysis revealed that exosomal CD40 achieved an AUC of 0.827 in distinguishing PDAC from clinical controls. Combining exo-CD40 along with exo-CD25 and CA19-9 discriminated PDAC patients from clinical controls with an AUC of 0.92. Exo-CD40 and exo-CD25 proteins found in exosomes isolated from plasma can serve as excellent non-invasive biomarkers for the early diagnosis of PDAC. Further larger scale studies are needed to validate combined exo-CD40 and exo-CD25 as a diagnostic tool for the identification of PDAC patients through non-invasive liquid biopsy.

## 1. Introduction

Pancreatic cancer stands out as one of the deadliest types of cancer. According to pancreasfoundation.org, pancreatic ductal adenocarcinoma (PDAC) ranks as the fourth leading cause of cancer-related deaths in the United States. Recent estimates from the American Cancer Society projected that in 2021, around 60,430 Americans would receive a pancreatic cancer diagnosis, with over 44,050 losing their lives to the disease [1]. Similarly, data from the Zentrum für Krebsregisterdaten in 2020 revealed that about 20,230 individuals in Germany were diagnosed with pancreatic cancer, resulting in a similar number of fatalities due to its poor prognosis [2]. A significant challenge associated with pancreatic cancer lies in its stealthy nature, with symptoms typically not appearing until the disease has progressed to advanced stages. By this point, the cancer cells have often spread to other areas of the body, making surgical removal of tumors difficult. Currently, the detection of PDAC relies on a combination of tests and techniques such as the CA19-9 antigen test, computerized tomography scans, and endoscopic biopsies [3,4]. The levels of CA19-9 and its sensitivity as a diagnostic marker improve with the increasing PDAC stage. However, in the early stages, the level of CA19-9 is similar to patients found in several benign conditions, precancerous lesions, and other cancers. This results in low specificity, as these methods can be invasive or lack specificity [5,6,7]. Research indicates that the development of pancreatic cancer from initial mutations to metastatic disease can take several years, underscoring the critical need for effective early detection methods [8]. As such, there is a pressing demand for the discovery and validation of new, sensitive, and non-invasive biomarkers for diagnosing PDAC. Enhancing early detection rates is vital in improving the overall survival rates of patients with pancreatic cancer.

In recent years, liquid biopsy has become increasingly popular as a safe, fast, and non-invasive alternative to traditional tissue biopsy [9,10]. However, to date, no single biomarker or multi-marker panel can reliably diagnose PDAC, especially in the early stages. Moreover, several groups have highlighted the importance of multi-biomarker panels for the early diagnosis of pancreatic cancer [11]. One type of liquid biopsy is focused on exosomes, which are a subset of extracellular vesicles (EVs) released by all cells and typically ranging in size from 30 to 150 nm [12,13]. Exosomes are known for their diverse nature, mirroring the characteristics and functions of their parent cells [14,15]. These EVs act as carriers of proteins, lipids, and nucleic acids between cells, making them crucial for intercellular communication. Exosomes are particularly rich in tetraspanin proteins like CD9, CD63, and CD81, which play a vital role in their formation, binding, fusion, and targeting, as well as their uptake by recipient cells [16,17]. The proteins, lipids, and nucleic acids carried by exosomes have shown promise as reliable biomarkers for cancer detection and prognosis. They also hold potential for innovative cancer treatments [18]. Various studies have highlighted the involvement of exosomes in cancer progression, metastasis, and resistance to chemotherapy [19,20]. Given these findings, exosomes have emerged as valuable non-invasive biomarkers for the early detection of pancreatic ductal adenocarcinoma (PDAC) or as targets for therapeutic interventions in PDAC treatment.

CD40 is a cell surface protein belonging to the tumor necrosis factor receptor superfamily, and it plays a vital role in the immune system. It is found on various immune cells, such as B cells, monocytes, and dendritic cells, as well as in a range of other cell types, including cancer cells, making it a new focus in cancer biology research [21,22,23,24]. CD40 has been utilized in several clinical trials as monotherapy or in combination with chemotherapy or mFOLFIRINOX (NCT03329950, NCT05650918). The administration of CD40 antibodies has resulted in significant and durable responses in PDAC patients [25]. Additionally, CD40 can be present on exosomes; its role on these vesicles has been linked to cancer. In the context of cancer, activating CD40 in immune cells associated with tumors can enhance the anti-tumor immune response [26]. In the case of pancreatic ductal adenocarcinoma (PDAC), which often exerts immunosuppressive effects, targeting CD40 has shown potential in altering the immune microenvironment of tumors.

CD25 is a protein found on the surface of immune cells, particularly on regulatory T cells (Tregs), where it is essential for immune regulation and tolerance [27]. Tregs, which are distinguished by high levels of CD25, play an important role in suppressing anti-tumor immune responses in pancreatic ductal adenocarcinoma (PDAC). As a primary marker of Tregs, CD25 aids in the inhibition of immune cells; the presence of Tregs is linked to a poorer prognosis, as it restricts the anti-tumor capabilities of the immune system.

In this study, we conducted a comprehensive analysis of 37 exosomal surface markers utilizing a MACS multiplex bead-based technique on plasma samples from healthy individuals, pancreatitis patients, and PDAC patients. Our goal is to identify key exosomal proteins that could potentially aid in distinguishing PDAC patients from both clinical control individuals and those with pancreatitis.

## 2. Results

### 2.1. Isolation and Characterization of Plasma-Derived Exosomes from Pancreatic Ductal Adenocarcinoma (PDAC) Patients

EVs from clinical control, pancreatitis, and PDAC patients were isolated using ultracentrifugation methods (Figure 1A), and called exosomes based on the following measurements and observation. Immunogold labeling and TEM determined the presence of CD9 on the isolated exosomes (Figure 1B). The NanoSight^®^ (Salisbury, UK) nanoparticle tracking analysis (NTA) revealed the diameter and concentration, respectively (Figure 1C,D). As the exosomes from pancreatitis patients exhibited a larger size than clinical control and PDAC patients, we assumed it could be due to the difference in lipid proportion of our patient’s cohort. We determined high-density lipoprotein (triglycerides) and low-density lipoprotein (Cholesterin) in our patient’s cohort. We did not see any difference in the lipid proportions in our patient’s cohort (Appendix A). The BCA protein assay measurement showed no significant differences between the exosome protein concentration of clinical controls, pancreatitis, and PDAC samples (Figure 1E). The Western blot analysis showed that our EVs from the clinical control, pancreatitis, and PDAC patients were positive for CD63 and CD9 (Figure 1F). We also included Calnexin as a negative exosomal marker, as it is expressed only in cell extract and not in exosomes (Appendix A). In summary, these results verify the purity of plasma-derived exosomes used in this study.

The FACS analysis of exosomes from clinical control, pancreatitis, and PDAC patients using the Miltenyi MACSPlex kit IO, human (Cat #130-108-813, 51429, Bergisch Gladbach, Germany) involved the use of 39 hard-dyed capture bead populations, each coated with different monoclonal antibodies against 37 potential extracellular vesicle (EV) surface antigens. These antigens included tetraspanins, cell adhesion molecules, immunoglobulins, platelet and endothelial cells, B cells, T cells, stemness, and proliferation markers, as well as two internal isotype negative controls. The bead populations were characterized and gated based on their fluorescence intensity as per the manufacturer’s instructions (Appendix A). Following incubation with EVs, the bulk bead-captured EVs were detected by counterstaining with APC-labeled detection antibodies against the tetraspanins CD9, CD63, and CD81, which are common EV surface markers. The analysis revealed several markers that showed a significant increase in the surface of EVs (Appendix A), including HLA-ABC, HLA-DRDPDQ, and CD31 (Appendix A). Appendix A represents the median of the 37 screened biomarkers in clinical control, pancreatitis, and PDAC patients. However, the markers that demonstrated a significant distinction between PDAC, pancreatitis, and clinical control groups were exo-CD40 and exo-CD25. These findings suggest that exo-CD40 and exo-CD25 could serve as potential biomarkers for distinguishing between PDAC, pancreatitis, and clinical control individuals based on the surface protein expression of EVs.

### 2.2. Exosomal CD40 Discriminates Pancreatic Ductal Adenocarcinoma (PDAC) and Pancreatitis Patients from Clinical Control Individuals

Exo-CD40 showed a significant difference in their relative number between the different sample groups. The expression of CD40 was significantly decreased in PDAC (n = 48, **** *p* < 0.0001) and pancreatitis (n = 21, *** *p* < 0.0002) patients in comparison to clinical control individuals (n = 51); however, no difference in the expression of CD40 was observed between pancreatitis and PDAC patients (Figure 2A). The clinicopathological parameters pTNM, perineural invasion, and tumor grading stage are linked to the progression of pancreatic cancer. Therefore, a detailed correlation of exosomal CD40 with different pathological parameters was performed. No difference in the expression of CD40 in PDAC patients was observed with respect to the nodal category (pN0 vs. pN+) (Figure 2B), perineural invasion (Pn0 vs. Pn1) (Figure 2C), tumor category (pT1/2 vs. pT3/4), metastasis (M0 vs. M1), grading (G1/2 vs. G3/4), and neoadjuvant treatment (yes vs. no) (Appendix A). To determine the prognostic relevance of exosomal CD40, PDAC patients were grouped into low (CD40 > median) and high (CD40 < median) to correlate it to overall survival. Appendix A presents the characteristic features of the study population divided based on the median exo-CD40 levels. We found that patients with a lower expression of exo-CD40 had a significantly favorable overall survival rate in comparison to the higher exo-CD40 expression (Figure 2D). As age is one of the important factors during pancreatic cancer, we analyzed whether age of the PDAC, pancreatitis or clinical control patients played a role in the expression of exo-CD40. We did not see any alterations (Appendix A).

In order to ascertain the significance of exo-CD40 in distinguishing PDAC, pancreatitis patients and clinical control individuals, we evaluated it by performing plasma CD40 ELISA. Our data showed no significant differences between patient groups (control vs. PDAC) (Appendix A). As CD40, being part of the TNF super family and one of the important proteins revealing the activation status of B cells, we analyzed the TNF levels (Appendix A), expression of CD40 on leucocytes (CD45+), B cells (CD20+) and monocytes (CD14) of clinical control and PDAC patients. No alterations in the expression of CD40 were found either on memory or naïve B cells in the circulation of clinical control and PDAC patients (Appendix A). We also determined the total number of B cells in both control and PDAC patients, observing a notable increase in PDAC patients compared to the clinical controls (Appendix A). In summary, these findings underscore the significance of exo-CD40 in distinguishing PDAC and pancreatitis patients from clinical control individuals.

### 2.3. Exosomal CD25 as a Marker to Discriminate Pancreatic Ductal Adenocarcinoma (PDAC) Patients from Pancreatitis Patients and Clinical Control Individuals

Next, we explored the exo-CD25 expression in clinical control, pancreatitis, and PDAC patients. We found a significant decrease in PDAC patients in comparison to pancreatitis patients (* *p* < 0.034) and control individuals (** *p* < 0.001). However, no difference in the expression of CD25 was observed between pancreatitis patients and control individuals (Figure 3A). Next, we analyzed if a different pathological extent influenced the expression of exo-CD25. It indeed influenced the nodal category and perineural invasion status of PDAC patients. PDAC patients with positive nodal category (pN1/pN2) had a significantly lower expression than PDAC patients with negative nodal category (pN0) (Figure 3B). PDAC patients with perineural invasion reflected findings opposite to the nodal category. Here, PDAC patients with perineural invasion (Pn1) status had a significantly higher exo-CD25 expression than PDAC patients without perineural invasion (Pn0) (Figure 3C). No differences were found in other pathological examinations like tumor category (pT1/pT2 vs. pT3/pT4), metastasis, grading and treated (neoadjuvant) against untreated PDAC patients (Appendix A). Further, to determine the influence of exo-CD25 in the survival of PDAC patients, we grouped PDAC patients into low and high groups according to their median expression. PDAC patients with higher exo-CD25 had reduced overall survival when compared to PDAC patients with low exo-CD25 expression (Figure 3D). Characteristic features of the study population dichotomized by the median exosomal CD25 are listed in Appendix A. Similar to exo-CD40, age did not influence the expression of exo-CD25 in clinical control, pancreatitis, and PDAC patients (Appendix A).

In order to compare with exo-CD25, we also examined the plasma concentration of CD25 in both clinical control subjects and patients with PDAC. Soluble CD25 levels of clinical control and PDAC patients showed no difference in their measurement (Appendix A). As CD25 is one of the activation markers of T cells, we analyzed its expression on leucocytes, T cells, helper T cells, and cytotoxic T cells. We found no alteration in the expression of CD25 between clinical control and PDAC patients of any of the cell types mentioned above (Appendix A). Given that CD25 is expressed on Tregs, we examined the correlation between exo-CD25 levels and the frequencies of Tregs (CD4+FoxP3+). However, we did not observe any correlation between these factors (Appendix A). Together, this finding reveals the importance of exo-CD25 in discriminating clinical control individuals from PDAC patients as well as pancreatitis patients from PDAC patients.

### 2.4. Validation of Exosomal CD40, CD25, and Serum CA19-9 as Combinatory Marker for the Diagnosis and Prognosis of Patients with Pancreatic Ductal Adenocarcinoma

The early detection of pancreatic cancer remains challenging due to the nonspecific symptoms of the disease, and the existing biomarkers, such as CA19-9, are not reliable on their own to detect early-stage pancreatic cancer. Previous findings indicate that individuals suffering from acute pancreatitis often present increased levels of CA19-9, which signifies that this marker lacks specificity [28]. Here, we explored the potential of an exosome-based liquid biopsy to detect cancer at an early stage. We focused on investigating the diagnostic utility of exo-CD40 and exo-CD25 as a novel marker for PDAC with plasma samples from patients with PDAC, pancreatitis, and clinical control individuals. We next measured serum CA19-9 levels, a circulating protein currently used in clinics as a tumor biomarker for screening and management of patients with pancreatic ductal adenocarcinoma. We found increased levels of CA19-9 in PDAC patients in comparison to pancreatitis patients and clinical control individuals. CA19-9 levels discriminated PDAC patients from pancreatitis patients (*p* = 0.0013) and PDAC patients from clinical control individuals (*p* ≤ 0.0001), but did not have any influence on the pTNM status, survival, and age of the PDAC patients (Figure 4A–D). Characteristics of the study population grouped according to the median CA19-9 level are shown in Appendix A.

To evaluate the predictive power of the exosomal biomarkers CD40, CD25, and CA19-9 individually or in combination, a generalized linear model was used to discriminate PDAC patients from clinical controls and PDAC patients from pancreatitis patients. The optimal cut-off values were estimated by maximizing the sum of sensitivity and specificity for different analyses. The best results of each analysis are reported for their optimal cut-off values and their corresponding sensitivity, specificity, and AUC values (Figure 5A). We demonstrate that the combination of biomarkers exo-CD40+, exo-CD25+, and CA19-9 produced the highest AUC value of 0.92, effectively distinguishing PDAC patients from clinical control subjects (Figure 5B). In comparison, the combinations exo-CD40+exo-CD25, exo-CD25+CA19-9, and exo-CD40+CA19-9 yielded AUC values of 0.8647, 0.8187, and 0.8948, respectively. Additionally, the effectiveness of each individual biomarker is detailed in Figure 5B, which includes the optimal cut-off values. The ROC curves for various biomarkers in patients with pancreatic ductal adenocarcinoma (PDAC) and pancreatitis are illustrated in Figure 5C.

Additionally, Figure 5D presents the comparative assessment of different biomarkers for PDAC patients versus chronic pancreatitis (Figure 5C). The results reported in the table indicate that the combination of biomarkers exo-CD40+exo-CD25+CA19-9 has the highest predictive power with an AUC value of 0.841 (Figure 5D). The performance of other combinations such as exo-CD40+exo-CD25, exo-CD25+CA19-9, and exo-CD40+CA19-9 and individual markers are shown in the tables (Figure 5B,D).

As patients with high CA19-9 levels are considered high risk for PDAC patients, we next identified PDAC patients with low CA19-9 and compared their exo-CD40 and exo-CD25 expression with the clinical control patients. We observed a significant difference in the expression of exo-CD40 and exo-CD25 between clinical controls and PDAC patients. Indeed, this finding would assist in decision making for PDAC patients. However, we need a larger sample volume to confirm our findings before application in clinical settings (Appendix A). Together, our findings report the predictive capability of the exosomal biomarkers in combination with serum CA19-9.

## 3. Discussion

Pancreatic cancer is one of the deadliest types of cancer due to the fact that most patients are only diagnosed at a very late stage when the cancer has already spread. This results in a very poor prognosis, with most patients succumbing to the disease within a year of diagnosis [29,30,31]. Therefore, it is crucial to identify new biomarkers that can help detect pancreatic cancer at an earlier stage. In this study, we looked into the potential of using exosome-based liquid biopsies as a method for the early detection of pancreatic cancer. Exosomes are small vesicles that are released into the bloodstream by both healthy and cancerous cells, carrying molecular information that can be used for communication between cells [32,33]. By analyzing these exosomes in the blood, we can gain important insights into the changes in cellular activity caused by cancer.

Exosomes have been shown to play a crucial role in immune response and tumor development, making them a promising target for cancer detection. In this study, we examined exosomes in the plasma of patients with pancreatic cancer, pancreatitis, and healthy individuals to see if there are any differences that could be used as biomarkers for detecting pancreatic cancer specifically. The approach of MACSPlex exosome assay was used in the study, and 37 proteins were analyzed, and we found the amount of CD40 surface protein on exosomes released in the plasma of PDAC patients was significantly lower than that in the plasma of clinical control individuals.

No change was detected in the exo-CD40 expression levels when compared with different pathological factors. Surprisingly, it was observed that patients with pancreatic ductal adenocarcinoma (PDAC) who had low exo-CD40 expression had a longer survival rate compared to those with high exo-CD40 expression. However, these results did not align with the exo-CD40 expression in clinical controls as opposed to patients with PDAC and pancreatitis. The reduced CD40 signal on plasma exosomes from PDAC and pancreatitis patients compared to those from clinical controls could be due to lower levels of circulating dendritic cells (DCs) and a significant increase in the absolute number of B cells. The role of B cells in PDAC is not well understood, with some studies suggesting a beneficial role while others indicating a detrimental one [34,35,36,37,38,39]. Our data suggest a decrease in exo-CD40 expression could indicate tumor progression in PDAC. Previous research has shown that systemic inflammation is linked to poor outcomes in PDAC patients. CD40 is known to be involved in the inflammatory pathway and its impact on PDAC patient outcomes [40]. CD40 is being investigated as a promising therapeutic target for pancreatic cancer. Numerous clinical trials have centered on CD40 through the use of agonists or synergistic approaches. One clinical study demonstrated the efficacy of the CD40 agonistic monoclonal antibody APX005M (sotigalimab) in combination with chemotherapy during a multicenter, phase 1b trial aimed at treating metastatic pancreatic adenocarcinoma, assessing safety, and determining the appropriate dose for phase 2 [41]. Additionally, research has indicated that CD40 agonists can enhance the effectiveness of immune checkpoint inhibitors, such as anti-PD1 and anti-CTLA4, in preclinical models of pancreatic ductal adenocarcinoma (PDAC). Clinical trials of these agents have shown potential in increasing T cell infiltration in tumors and enhancing survival rates when used alongside chemotherapy or other immunotherapies [42]. The combination of CD40 agonists with other therapies, such as chemotherapy (e.g., gemcitabine) or immune checkpoint inhibitors, is actively being investigated. The rationale behind this combination is that chemotherapy can reduce tumor load and potentially expose more tumor antigens, while CD40 agonists may strengthen the immune system’s ability to detect and target these antigens [43,44,45]. Despite the promising potential of CD40 activation, several challenges persist. In certain cancers, tumor cells may produce CD40L, which can hinder the desired immune response. The expression of CD40L on tumor cells could diminish the effectiveness of CD40-targeted treatments or even promote tumor advancement. Additionally, CD40L is found on various immune cells, and its activation in macrophages or dendritic cells may lead to inflammation and harm to healthy tissues [46]. Therefore, it is essential to manage its activation carefully to prevent excessive immune stimulation and any related toxic effects.

Furthermore, there was a notable difference in exosomal CD25 expression between patients with pancreatic ductal adenocarcinoma (PDAC) and those with pancreatitis, as well as clinical control individuals. Low levels of exo-CD25 in PDAC patients were closely associated with the nodal category. Specifically, patients with positive regional lymph nodes exhibited reduced exo-CD25 levels compared to those without such infiltration. On the other hand, the status of perineural invasion in PDAC patients showed opposite findings regarding exo-CD25 expression. Patients with perineural invasion displayed higher levels of exo-CD25 compared to those without this invasion. CD25 acts as an activation marker for T cells and is typically found on regulatory T cells. These cells tend to accumulate in the tumor microenvironment of pancreatic ductal adenocarcinoma (PDAC), where they suppress the immune response and help tumor cells evade detection and destruction by the immune system. Numerous factors contribute to the poor prognosis of patients with PDAC, one of which is perineural invasion, characterized by significant interactions between tumor cells, nerves, and the tumor microenvironment. In our study, we demonstrate that patients with PDAC showing perineural invasion (Pn1) had notably higher levels of exo-CD25 compared to those without this invasion (Pn0). This observation corresponds with earlier studies indicating that elevated levels of CD25 are linked to worse outcomes in PDAC patients. Furthermore, murine models of pancreatic cancer have shown that the depletion of CD25 leads to decreased tumor burden and enhanced survival rates [47,48,49]. Research has indicated that elevated levels of CD25 on Tregs in PDAC tumors are frequently linked to unfavorable patient prognoses, and increased Treg infiltration (including CD25+ cells) in the tumor microenvironment is associated with lower survival rates [50]. Additionally, a separate study explored the use of anti-CD25 treatments alongside 5-FU for managing PDAC [51]. These results strongly suggest that heightened exo-CD25 expression and perineural invasion status may influence the survival outcomes of patients with PDAC. Notably, patients with lower exo-CD25 expression tended to have longer survival periods compared to those with elevated exo-CD25 levels. Interestingly, exo-CD25 expression levels could effectively distinguish between PDAC patients and those with pancreatitis. In conclusion, the results regarding exo-CD25 expression and its association with perineural invasion highlight its potential as a valuable marker for distinguishing PDAC patients from both clinical controls and patients with pancreatitis.

This study aimed to assess the predictive ability of exosomal markers compared to the traditional serological tumor marker CA19-9, which is commonly used for diagnosing PDAC. While CA19-9 has been shown to be useful for managing pancreatic cancer, its effectiveness in early detection is limited. In cancer screening, it is crucial to have high sensitivity and specificity to accurately identify the disease. Our study found that using CA19-9 alone had a moderate AUC of 0.76 in distinguishing PDAC patients from clinical controls. However, when combined with exo-CD40 (AUC-0.89), exo-CD25 (AUC-0.81), or both exosomal markers, the final predictive model had an AUC of 0.92, significantly better than CA19-9 alone. This combination also outperformed CA19-9 in distinguishing PDAC from pancreatitis patients. These results suggest that incorporating exosomal markers alongside CA19-9 could enhance the diagnostic accuracy and could offer a more comprehensive profile for pancreatic cancer early detection and treatment monitoring.

Although CA19-9 is currently the most important biomarker for detecting pancreatic cancer, a significant percentage of patients with PDAC do not show elevated levels of CA19-9 [52,53,54]. Additionally, some patients do not produce the Lewis antigen, resulting in low or no secretion of CA19-9 [55]. This limitation has led to the recognition that CA19-9 alone may not be sufficient for the accurate diagnosis of PDAC. Given the shortcomings of CA19-9 in diagnosing PDAC, it is critical to explore other biomarkers that can complement its use, especially in cases of Lewis-negative pancreatic cancer. Our research revealed that a notable percentage (24.48%) of PDAC patients had CA19-9 levels below the normal threshold of 37 U/mL, leading to a misdiagnosis of pancreatic cancer. A subsequent analysis within the group of PDAC patients with normal CA19-9 levels demonstrated that exo-CD40 showed promise in distinguishing patients with PDAC. In conclusion, our findings emphasize the value of exosomal expression as a reliable indicator of PDAC, particularly in cases where CA19-9 levels are not elevated. This underscores the importance of incorporating additional biomarkers alongside CA19-9 in the screening and diagnosis of PDAC patients.

Chemotherapy, alongside surgery and radiation, is a primary treatment for cancer, but it has been shown to enhance the secretion of extracellular vesicles (EVs) by cancer cells, which can modulate immune responses and alter the composition of those EVs, thereby potentially increasing the cancer cells’ adaptability to resist therapy [56,57]. Research has demonstrated that extracellular vesicles (EVs) play a significant role in regulating immune responses, encompassing both immune activation and suppression [58,59]. Further studies indicate that chemotherapy can modify the contents of these released EVs, potentially impacting the cells that receive them. For example, research by Bandari et al. revealed that EVs induced by chemotherapy exhibited increased levels of heparanase on their surfaces [56]. In addition to proteins, EVs are known to transport nucleic acids, including microRNAs (miRNAs), messenger RNAs (mRNAs), and non-coding RNAs [60]. It has been found that breast cancer cells subjected to chemotherapy alter the composition of miRNA-laden EVs, such as miR-9-5p and miR-195-5p. These changes influence the transcription factor One Cut Homeobox 2, which ultimately affects the expression of stemness-related genes like NANOG, OCT4, and SOX2, contributing to the increased adaptiveness of cancer cells against therapies.

Our findings present compelling evidence supporting the potential clinical relevance of utilizing exosome-based biomarkers for a non-invasive liquid biopsy method to differentiate between PDAC patients, pancreatitis patients, and clinical controls.

However, it is important to acknowledge that our study is limited by a relatively small sample size, resulting in limited statistical power. Therefore, before exo-CD40 and exo-CD25 along with CA19-9 can be used in clinical practice, large-scale validation studies are needed to confirm its diagnostic and prognostic value in PDAC. Furthermore, we did not perform any in vivo or in vitro experiments to evaluate the functional significance of the identified exosome biomarkers. Additionally, we did not provide information regarding the source of exo-CD40 and exo-CD25.

## 4. Materials and Methods

### 4.1. Patients

This study included 120 samples gathered from patients who underwent surgery at the University Hospital Erlangen in Germany between 2020 and 2022. The study was conducted in accordance with the Declaration of Helsinki and approved by the Institutional Review Board of the University Hospital Erlangen (No. 180_19 B, 14.06.2019). Study patients or their legal guardians signed the written informed consent form prior to surgery. The patients were divided into three groups: clinical controls (n = 51), pancreatitis (n = 21), and PDAC (n = 48). Patients with a confirmed diagnosis of pancreatic ductal adenocarcinoma (PDAC) or pancreatitis through histopathological examination post-surgery were included in this study. Clinical controls were age matched with pancreatitis and PDAC patients (Table 1).

### 4.2. Sample Preparation

Peripheral blood (28 mL) was collected in an ethylene diaminetetraacetic acid (EDTA) coated tube (Cat-No. 04.1921.001, Sarstedt, Nürnbrecht, Germany) prior to surgery. Plasma samples were separated at a centrifugation of 350× *g* for 10 min without brake to remove dead cells and cell debris and stored at −80 °C until subsequent analysis.

### 4.3. Exosome Isolation

Exosome isolation was performed through subsequent centrifugation steps: Plasma samples (4 mL) were centrifuged at 300× *g* for 10 min (to eliminate cellular components), 2000× *g* for 30 min (to remove cellular debris), 10,000× *g* for 30 min (removal of bigger EVs). For the enrichment of exosomes, two ultracentrifugation steps at 100,000× *g* for 2 h were performed. Plasma EVs were filtered through Millipore Express PLUS Membrane, Polyethersulfon, hydrophil, 0.22 µm, 13 mm (Catalog No. GPWP01300, Merck, Darmstadt, Germany) in between the two ultracentrifugation steps. EV pellets were resuspended in 500 µL of PBS (Figure 1A). The EV concentration was measured by the Pierce BCA protein assay Kit (Catalog No. 23227, Thermo Scientific, Waltham, MA, USA) using BSA standards (Figure 1).

### 4.4. Bicinchoninic Assay (BCA) Protein Assay

The EV and respective exosome concentration were measured by the Pierce BCA protein assay Kit (Catalog No. 23227, Thermo Scientific) using bovine serum albumin (BSA) standards. The kit was used according to the manufacturer’s recommendations.

### 4.5. Nanoparticle Tracking Analysis

The particle size distribution in the purified extracellular vesicles (EVs) was determined using a Nanoparticle Tracking Analysis (NTA) instrument, specifically a Zetaview PMX-110 instrument (Particle Metrics, Inning am Ammersee, Germany) equipped with a 405-nm laser. Prior to sample measurement, the instrument was calibrated using 100 nm polystyrene beads diluted in water as per the manufacturer’s instructions. The measurements were conducted at a constant cell temperature of 25 °C. Samples were appropriately diluted in PBS to a total volume of 1 mL for measurement. Each measurement cycle involved scanning eleven cell positions, with video recording at 30 frames per second. Additional capture settings included a gain of 719.52, shutter speed of 50, and a minimum trace length of 15. The recorded videos were analyzed using ZetaView software version 8.05.12, with specific settings such as minimum brightness of 25, maximum brightness of 255, minimum area of 5, and maximum area of 200. Finally, the concentration of EVs was calculated taking into account the appropriate dilution factors based on the manufacturer’s recommendations. This detailed methodology ensures the accurate and reliable determination of EV concentration and size distribution for research and analysis purposes.

### 4.6. Immunogold Labeling and Electron Microscopy

Fixed EV specimens (4% PFA in PBS mixed 1:1 with EV) were placed onto 10 min UV irradiated 300 mesh formvar/carbon coated grids and allowed to absorb to the formvar for 5 min. For immunogold staining, the grids were placed into 20 µL 0.01% Tween/PBS (10 min) and after that into a blocking buffer (0.5% fish gelatin with 0.1% ovalbumin in PBS) for a block step for 1 h. Without rinsing, the grids were immediately placed into the primary antibody (diluted in a blocking buffer) at the appropriate dilution overnight at 4 °C (1:100 anti-CD9 Abcam (Cambridge, UK), ab236630). As controls, some of the grids were not exposed to the primary antibody. The next day, all the grids were rinsed with PBS then floated on drops of the appropriate secondary antibody attached with 10 nm gold particles (AURION 1:30) for 2 h at room temperature (RT). Grids were rinsed 3 times with PBS and were placed in 1% glutaraldehyde (in PBS) for 5 min. After rinsing in PBS and distilled water, the grids were stained for contrast using 2% uranyl oxalate solution (pH 7 for 5 min in the dark). Afterwards, the grids were incubated in drops of methyl cellulose-uranyl oxalate (8 parts 2% methyl cellulose, 1 part ddH2O, 1 part 4% uranyl acetate (in water), pH 4, sterile filter) for 10 min on ice (dark) [61]. After that, the grids were removed with stainless steel loops and excess fluid was blotted by gently pushing on Whatman filter paper. After air-drying, the samples were examined and photographed with a Zeiss EM10 electron microscope (Zeiss, Jena, Germany) and a Gatan SC1000 Orius™ CCD camera (GATAN, Munich, Germany) in combination with the DigitalMicrograph™ software 3.1 (GATAN, Pleasanton, CA, USA). Images were adjusted for contrast and brightness using Adobe Photoshop CC 2018 (Adobe Systems, San José, CA, USA).

### 4.7. MACSPlex Exosome Assay

The screening assay (Catalog No. 130-108-813, MACSPlex Human Exosome Kit; Miltenyi, Bergisch Gladbach, Germany) was previously described [62,63]. In brief, the assay is based on 4–8 µm diameter poly-styrene beads, labeled with different amounts of 2 dyes (phycoerythrin and fluorescein isothiocyanate), to generate 39 different bead subsets recognized by flow cytometry analysis. Each bead subset is conjugated with a different capture anti-body that recognizes EVs carrying the respective antigen (37 EV surface epitopes plus 2 isotype controls). Beads were incubated with a sample overnight. Next day EVs bound to beads were detected by allophycocyanin-conjugated anti-CD9, anti-CD63, and anti-CD81 antibodies (Appendix A). Finally, plasma samples were analyzed with the special order research product (with blue, red, violet, UV, YellGrn laser). PBS was used to measure the background signal. The median fluorescence intensity (MFI) of each EV marker was normalized to the mean MFI for specific EV markers (CD9, CD63, and CD81). For the calculation of a relative number of exosome surface markers, first the median signal intensity of each bead obtained from the buffer as a control sample was subtracted from the signal intensities of the respective beads incubated with the sample. Finally, the signal intensities of all beads were divided by the normalization factor of the respective sample. The mean of the median signal intensity of the beads CD9, CD63, and CD81 was used as the normalization factor for each sample.

### 4.8. Flow Cytometry

For peripheral blood analysis, samples were lysed with a lysis buffer. Cells were stained with antibody cocktail for 30 min at 4 °C. Afterwards, the cells were washed with FACS buffer (1X PBS, FCS (1%), BSA (0.5%), EDTA (2 mM)) and were ready to be acquired. The antibodies were anti-CD45 BV786 (clone-HI30, BD (#563716)), anti-CD20 BV650 (clone-2H7, BD (#563780)), anti-CD40 BUV737 (clone-5C3, BD (#741847)), anti-CD27 BV711 (clone-L128, BD (#563167)), anti-IgD Per/Cp (clone-IA6-2, Biolegend (#348234)), anti-CD3 BUV395 (clone-SK7, BD (#564000)), anti-CD4 Per/Cp-Cy5.5 (clone-RPA-T4, BD (#560650)), anti-CD8 PE Dazzle/CF 594, (clone-RPA-T4, BD (#562311)), anti-CD25 BV711 (clone- M-A251, BD (#740776)), anti-FoxP3 PE (clone-259D/C7, BD (#560046), anti-IgG1 PE (clone-MOPC-21, BD (#555748), anti-IgG1, k Isotype Control BUV737 (clone-X40, BD (#4046975), and anti-IgG1, k Isotype Control BV711 (clone-X40, BD #563044). Data were acquired on FACS Celesta (BD Biosciences, Franklin Lakes, NJ, USA) using the BD FACSDiVa™ software v8.0.1.1 and analyzed with FlowJo 10.3.0 (FlowJo LLC, Ashland, OR, USA).

### 4.9. Enzyme Linked Immunosorbent Assay (ELISA) of Human Cancer Antigen CA19-9, CD40, and CD25

Serum CA19-9, CD40 and CD25 protein levels in analyzed groups were assessed using the Cancer Antigen CA19-9 Human ELISA Kit (Catalog No. ab108642, Abcam), Human CD40 Quantikine ELISA Kit (Catalog No. DCCD40 R&D systems, Minneapolis, MN, USA) and Human CD25/IL-2R alpha Quantikine ELISA Kit (Catalog No. DR2A00 R&D systems), according to the manufacturer’s protocol.

### 4.10. Western Blot

Equal amounts of tissue lysates (10 µg protein) and equal volumes of EV samples (20 µL) were suspended in a sample buffer (0.05 M Tris–HCl (pH 6.8), 10% glycerol, 2% SDS, 1% bromophenol blue) under reducing (for CD40 staining; 1% beta-mercaptoethanol and 1.5% DTT added to a sample buffer) or non-reducing (for CD9 and CD63 staining) conditions, and boiled for 5 min at 95 °C. Proteins were separated by SDS-PAGE (SDS polyacrylamide gel electrophoresis), transferred to nitrocellulose membranes, blocked in 5% non-fat milk in PBS with 0.5% Tween-20, and immunostained overnight at 4 °C using the following primary antibodies in a 1/1000 dilution in TBST: CD9 (BD555370), CD63 (BD556019), CD40 (Cat. #66965-1-Ig, Proteintech, Rosemont, IL 60018, USA). Blots were developed using the SuperSignal West Femto reagent (Thermo Fisher, Waltham, MA, USA) and visualized on an Amersham 600 system (GE Healthcare, Chicago, IL, USA).

### 4.11. Measurement of Triglycerides and Cholesterin

Triglyceride (OSR 60118) and Cholesterin (OSR 6116) were measured using Beckman Coulter DxC 700 AU or AU 5800 (Brea, CA, USA) according to the manufacturer’s instructions.

### 4.12. Statistical Analysis

Statistical analyses were performed using GraphPad PRISM version 10 and IBM SPSS version 28. The variable distribution was identified by the Anderson–Darling test, D’Agostin and Pearson test, Shapiro–Wilk test, and Kolmogorov–Smirnov test. The correlation of normally distributed variables was analyzed by Pearson correlation coefficients, whereas the non-normally distributed parameters were analyzed by non-parametric Spearman correlation. The column analysis for normally distributed data was conducted by the unpaired *t* test with Welch’s correction. The non-normally distributed data were analyzed by the Kruskal–Wallis test. The receiver operating characteristic (ROC) analysis for the following experiment was performed using the ROCit R package (ROCit version 2.1.2). The Generalized Linear Model (GLM) was used for model training. It was applied on two data sets, where the first data set contains 49 PDAC samples and 19 pancreatitis samples and the second data set comprise 49 PDAC samples and 51 clinical control samples. Both of the data sets have exosomal biomarkers as a feature, namely, CD40, CD25, and CA19-9. The predictive capability for each exosomal biomarker was determined either individually or in combination. On each data set at first, the GLM model was applied and then the performance metrics were computed similar to ROC curves, area under curve (AUC), sensitivity, and specificity.

## 5. Conclusions

Our findings represent a novel and promising biomarker combination for the diagnosis and prognosis of PDAC. It provides strong evidence for the potential clinical significance of using exosome-based biomarkers as a non-invasive liquid biopsy method to differentiate between patients with PDAC, those with pancreatitis, and clinical controls. However, more research is needed to fully validate its clinical utility, and it may eventually be used in combination with other biomarkers for a more accurate and comprehensive evaluation of PDAC patients.

## Figures and Tables

**Figure 1 ijms-26-01500-f001:**
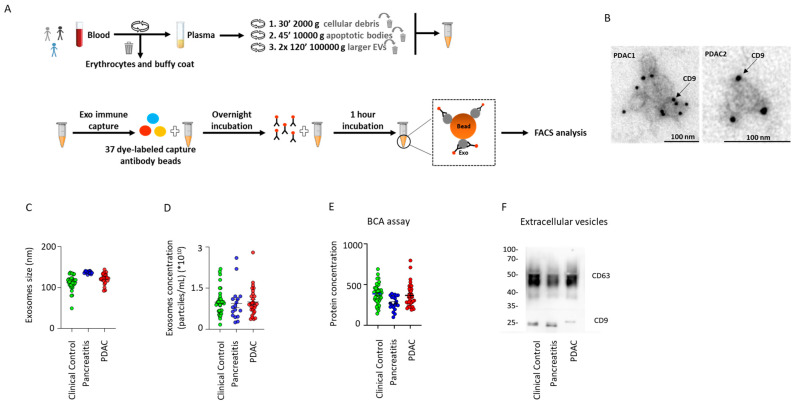
**Experimental overview of exosome enrichment and characterization.** (**A**) Schematic representation of exosome isolation from peripheral blood and MACSPlex exosome assay. (**B**) Transmission Electron Microscopy (TEM) images showing enriched plasma exosomes and exosome marker CD9 from two PDAC patients. (**C**,**D**) Measurement of nanoparticle concentration (N/mL plasma) by the nanoparticle tracking analysis (NTA) in controls (n = 36), pancreatitis (n = 16), and PDAC groups (n = 32). (**E**) Exosomal protein concentration, measured by the BCA assay, in controls (n = 43), pancreatitis (n = 20), and PDAC groups (n = 35). (**F**) Measurement of protein concentration by Western blot. Data are mean ± s.d.

**Figure 2 ijms-26-01500-f002:**
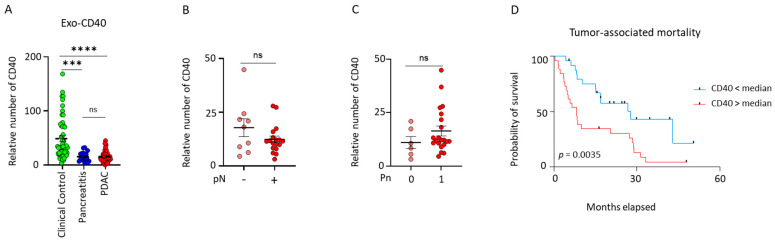
**Validation of exosomal CD40 as a novel biomarker for pancreatic cancer and pancreatitis.** (**A**) Normalized signal intensities of exosomal surface protein CD40 in plasma exosomes of controls (n = 51), patients with pancreatitis (n = 21), and patients with PDAC (n = 48) (Mann–Whitney test, *** *p* = 0.0010, **** *p* < 0.0001). (**B**,**C**) Correlations between normalized signal intensities of exosomal surface protein CD40 with nodal category and perineural invasion status in patients with PDAC. (**D**) Kaplan–Meier curves displaying survival analysis of patients with PDAC (Gehan–Breslow–Wilcoxon). Data are mean ± s.d.

**Figure 3 ijms-26-01500-f003:**
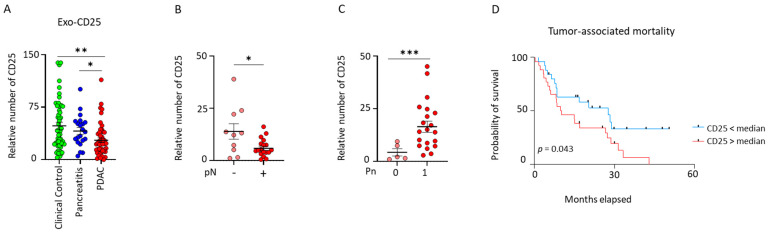
**Validation of exosomal CD25 as a novel biomarker for pancreatic cancer and pancreatitis.** (**A**) Normalized signal intensities of exosomal surface protein CD25 that differ between plasma exosomes of controls (n = 51), patients with pancreatitis (n = 21), and patients with PDAC (n = 48) (Mann–Whitney test, * *p* = 0.0104, ** *p* = 0.0026). (**B**,**C**) Correlations between normalized signal intensities of exosomal surface protein CD25 with nodal category and perineural invasion status in patients with PDAC (unpaired *t* test with Welch’s correction * *p* = 0.0364, *** *p* = 0.0010). (**D**) Kaplan–Meier curves displaying survival analysis of patients with PDAC (Gehan–Breslow–Wilcoxon). Data are mean ± s.d.

**Figure 4 ijms-26-01500-f004:**
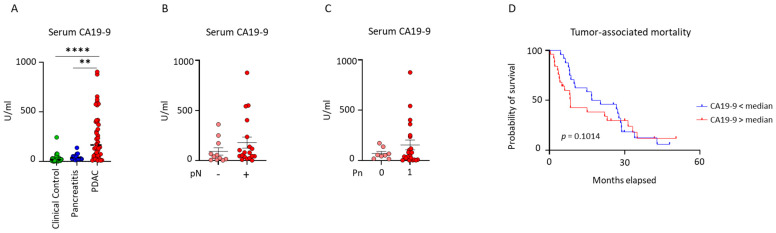
**CA19-9 to discriminate PDAC patients from clinical controls and pancreatitis patients.** (**A**) Serum CA19-9 levels in clinical control, pancreatitis, and PDAC patients Mann–Whitney test, ** *p* = 0.002, **** *p* = <0.0001. (**B**,**C**) Correlations between serum CA19-9 levels with nodal category and perineural invasion status in patients with PDAC. (**D**) Kaplan–Meier curves displaying survival analysis of patients with PDAC (Gehan–Breslow–Wilcoxon). Data are mean ± s.d.

**Figure 5 ijms-26-01500-f005:**
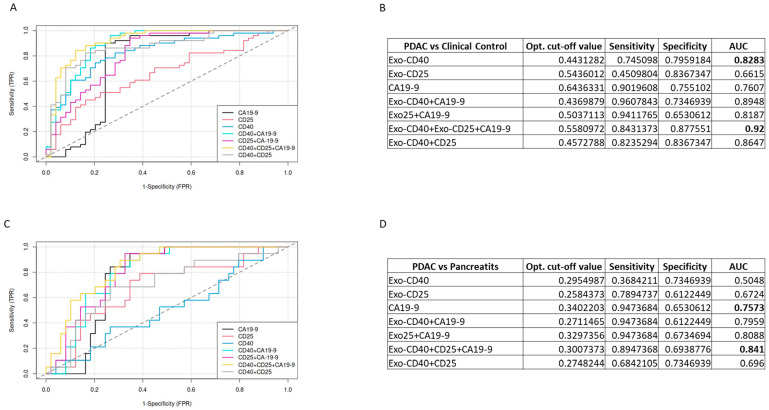
**Receiver operating characteristic (ROC) curves analysis of exosome-surface markers and CA19-9.** (**A**) ROC curves discriminating the PDAC group from clinical controls. Each ROC curve shows the single markers exo-CD40, exo-CD25, and CA19-9 and the combination with the highest AUC. (**B**) Table representing the cut-off values, sensitivity, specificity for exo-CD40, exo-CD25, and CA19-9 surface markers and the selected combination. (**C**) ROC curves discriminating the PDAC group from pancreatitis. (**D**) Table representing the cut-off values, sensitivity, specificity for exo-CD40, exo-CD25, and CA19-9 surface markers and the selected combination.

**Table 1 ijms-26-01500-t001:** Characteristic features of clinical control patients.

	Clinical Control Patients	Pancreatitis Patients	PDAC *p*-Value
Number	51	21	48
Mean Age (in years [range])	61.3 (38–86)	58.1 (43–78)	68.7 (51–86) <0.0001
Sex (Male:Female)	26:25	13:8	25:23 0.40
Performed surgery			
Cholecystectomy	12		
Acute ulceritis	8		
Hernie	5		
Thoraxmagen	2		
Healthy volunteers	24		

## Data Availability

The original contributions presented in this study are included in the article/Appendix A. Further inquiries can be directed to the corresponding author.

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
