# Peer review of "Exosomal CD40, CD25, and Serum CA19-9 as Combinatory Novel Liquid Biopsy Biomarker for the Diagnosis and Prognosis of Patients with Pancreatic Ductal Adenocarcinoma"

_ijms, 2025, doi:10.3390/ijms26041500_

Round 1

Reviewer 1 Report

Comments and Suggestions for Authors

The article by David et al proposed a potential biomarker panel of exosomal CD40, CD25 and CA19.9 for the early detection of PDAC. 37 exosomal proteins were screened in over 100 patients and controls. Conclusions were supported with sufficient data and discussions. Overall it is well written.

Here are some minor suggestions for consideration:

1. First about the format. The supplementary graphs were aligned to the right and cut-off when viewing. Please center the graphs in the middle.

2. For Figure SI-1, please add a ladder lane in the left panel of the picture and remove the 1-5 labels in that panel. Because it seems like the left panel only contains ladder, while all the samples are in the right panel.

3. For the manuscript part, in the Introduction section on page 2 line 57, the author mentioned the current diagnostics methods lack specificity. What about the sensitivity? What are the reported sensitivities?

4. The author found CD40 and CD25 as promising biomarkers out of 37 targets. There are clear plots showing the biomarker conc. differences between cancer patients and controls. It might also be useful if the author could summarize a table showing the average/median conc. of all 37 screened biomarkers in PDAC, pancreatitis and control.

5. What is the status of liquid biopsy biomarker discoveries in PDAC? Are there any reported potential biomarkers? Have CD40 and CD25 been reported as diagnostic or prognostic biomarkers for PDAC in other studies? Please include that information in the introduction section.

6. In section 2.1 on page 3 line 109, please explain a bit why choosing Calnexin as the negative control.

7. The author found that patients with positive regional lymph nodes showed decreased CD25 level, while patients with perineural invasion showed increased CD25 than those without. This seems to be related to CD25 functions in PDAC metastasis mechanisms. Just curious, are there various different ways in PDAC metastasis, e.g. lymph nodes vs. nerves, and how was CD25 involved in these different routes?

Author Response

We thank the reviewer for the interesting suggestions to improve the manuscript. Please find in attachment the point by point response. 

Reviewer 2 Report

Comments and Suggestions for Authors

In this manuscript, the authors present data demonstrating the following: (1) Exosomal CD40 is significantly lower in PDAC and pancreatitis patients compared to non-cancer controls, while exosomal CD25 is significantly lower in PDAC patients compared to both pancreatitis and non-cancer controls. (2) PDAC patients with high exosomal CD40 and CD25 levels have a worse prognosis compared to those with low exosomal CD40 and CD25 levels.(3) The combination of exosomal CD40+, exosomal CD25+, and serum CA19-9 achieved the highest AUC, effectively distinguishing PDAC patients from clinical controls and pancreatitis patients. Based on these findings, the authors propose exosomal CD40, exosomal CD25, and serum CA19-9 as a novel combinatory liquid biopsy biomarker panel for the diagnosis and prognosis of PDAC.

 1. Most diagnostic biomarkers for tumors are upregulated proteins. For example, in this study, CA19-9 is a commonly used serum biomarker for PDAC. However, its sensitivity remains suboptimal, as some PDAC patients do not exhibit elevated CA19-9 levels. The authors found that CD40 and CD25 were significantly downregulated in PDAC-derived exosomes. However, neither CD40 nor CD25 alone could effectively distinguish PDAC from clinical controls and pancreatitis, as some individuals in these groups exhibited similar expression levels. Compared to CA19-9, CD40 and CD25 individually do not offer a significant diagnostic advantage, but their combined detection improves PDAC prediction. Clinically, individuals with high CA19-9 levels are considered high-risk for PDAC. Thus, identifying PDAC in patients with low CA19-9 is of greater significance. It is recommended that the authors analyze the expression levels of CD40 and CD25 in exosomes from CA19-9-low PDAC patients and compare them to controls. If this subgroup can be identified through CD40 and CD25, it would have important clinical implications.

 2. An interesting phenomenon observed in this study is that exosomes from pancreatitis patients exhibit larger sizes but lower exosome concentration and protein content compared to those from control and PDAC patients. This suggests the possibility that exosomes derived from pancreatitis may contain a higher proportion of lipids. To further investigate this, the authors are encouraged to assess the lipid content of these exosomes. Identifying differences in total lipid content or specific lipid species could serve as a distinguishing feature between pancreatitis and PDAC, potentially offering a novel biomarker for differential diagnosis.

3. These issues should be addressed. The authors are advised to ensure consistency in labeling, particularly with regard to the figure references in parentheses. The font style (bold or not) should be uniform throughout. For figures where p-values can be calculated, the corresponding p-values should be included. Additionally, on line 239, the content described corresponds to Figure S5A-E, not Figure S4A-E. On line 269, the authors intended to refer to Figure 5D, but mistakenly wrote "the table (Figure 5D)." This should be corrected. 

Author Response

(The authors gave the same response as above.)

Round 2

Reviewer 2 Report

Comments and Suggestions for Authors

The authors have addressed my questions. Thank you.